# Age-dependent dormant resident progenitors are stimulated by injury to regenerate Purkinje neurons

N Sumru Bayin[1], Alexandre Wojcinski[1†], Aurelien Mourton[1‡], Hiromitsu Saito[2], Noboru Suzuki[2], Alexandra L Joyner[1,3]*

[1]Developmental Biology Program, Sloan Kettering Institute, New York, United States; [2]Department of Animal Functional Genomics of Advanced Science Research Promotion Center, Organization for the Promotion of Regional Innovation, Mie University, Tsu, JAPAN; [3]Biochemistry, Cell and Molecular Biology Program, Weill Cornell Graduate School of Medical Sciences, New York, United States

**Abstract** Outside of the neurogenic niches of the brain, postmitotic neurons have not been found to undergo efficient regeneration. We demonstrate that mouse Purkinje cells (PCs), which are born at midgestation and are crucial for development and function of cerebellar circuits, are rapidly and fully regenerated following their ablation at birth. New PCs are produced from immature FOXP2+ Purkinje cell precursors (iPCs) that are able to enter the cell cycle and support normal cerebellum development. The number of iPCs and their regenerative capacity, however, diminish soon after birth and consequently PCs are poorly replenished when ablated at postnatal day five. Nevertheless, the PC-depleted cerebella reach a normal size by increasing cell size, but scaling of neuron types is disrupted and cerebellar function is impaired. Our findings provide a new paradigm in the field of neuron regeneration by identifying a population of immature neurons that buffers against perinatal brain injury in a stage-dependent process.
DOI: https://doi.org/10.7554/eLife.39879.001

*For correspondence:
joynera@mskcc.org

Present address: †John Wayne Cancer Institute, Santa Monica, United States; ‡Center for Interdisciplinary Research in Biology, College de France, Paris, France

Competing interests: The authors declare that no competing interests exist.

## Introduction

Most neurons in the brain are generated at specific developmental time points, and once a neuron becomes postmitotic regeneration following injury is limited, except for in two forebrain regions that maintain neurogenesis (*Chaker et al., 2016*). In the context of injury, adult forebrain neurons undergo limited recovery that involves either reactive gliosis (*Buffo et al., 2008*; *Robel et al., 2011*; *Sirko et al., 2013*) or migration of neural stem cells from the neurogenic niches (*Benner et al., 2013*; *Llorens-Bobadilla et al., 2015*; *López-Juárez et al., 2013*; *Martí-Fàbregas et al., 2010*). The cerebellum (CB) of the hindbrain has a complex folded structure that houses the majority of neurons in the brain and is essential for balance and motor coordination, as well as higher order reasoning via circuits it forms throughout the forebrain (*Fatemi et al., 2012*; *Steinlin, 2007*; *Tavano et al., 2007*; *Tsai et al., 2012*; *Wagner et al., 2017*). For two weeks after birth, the postnatal mouse CB consists of both neurons generated in the embryo, and two neurogenic progenitor pools that produce late born neurons and glia. Interestingly, the proliferating granule cell progenitors can be replenished following injury by adaptive reprograming of the second Nestin-expressing progenitors (*Wojcinski et al., 2017*). However, once a neurogenic process has ended, the degree to which post mitotic neurons can undergo regeneration is poorly understood.

Purkinje cells (PC) are born by embryonic day (E) 13.5 in the mouse and during weeks 10–11 in humans (*Rakic and Sidman, 1970*; *Wang and Zoghbi, 2001*). After exiting the cell cycle in the ventricular zone, PCs express FOXP2 as they migrate to form a PC layer (PCL) under the cerebellar

**eLife digest** The cerebellum, or 'little brain', handles movement, thought and social interaction. Unlike the rest of the brain, which primarily develops in the womb, most of its cells appear within the first year of our lives (or first few weeks in mice). This makes it vulnerable to injury around the time of birth.

We used to think that the brain could not replace damaged neurons, but when specific precursor cells in the cerebellum in the brains of newborn mice are removed, they are able to renew themselves. This is because specialized stem cells start to divide and produce the missing cells of the cerebellum.

Another type of cells in the cerebellum, called Purkinje neurons, are already produced in the embryo. They direct the development of several other cell types in the cerebellum after birth. They are also a crucial component of the circuits within the cerebellum, and losing them can cause loss of muscle coordination. Purkinje cells do not normally divide once an animal is born, but scientists want to know if they might be able to regrow after injury at birth.

Bayin et al. killed Purkinje cells in newborn mice with a toxin and used fluorescent markers to track the dying cells. Then, the remaining cells in the surrounding area were studied. This revealed that even when half of the Purkinje cells died a day after birth, the mice behaved normally. The cells regrew, and the cerebellum developed as it should. This happened because the loss of the Purkinje cells activated a population of immature Purkinje cells (iPCs). These cells would normally mature into adult Purkinje cells, but in their immature state they can still divide and make copies of themselves to replace lost neurons after injury.

As the mice grew older, the number of iPCs started to drop as the immature cells developed into adult Purkinje cells. When the iPCs ran out, any cells available to divide were gone and the mice could no longer replace any damaged Purkinje cells – the repair window had closed.

This work raises the possibility that other types of immature cells in the brain could be set aside to help repair damage during early development. A better understanding of these cells could reveal clues about conditions such as autism, which have been linked to damages or faults in the cerebellum. It may also help to gain new insights into how to regenerate the adult brain after injury.
DOI: https://doi.org/10.7554/eLife.39879.002

surface by E17.5, and turn on Calbindin1 (CALB1) in the late embryo and stop expressing FOXP2 by two weeks after birth. PCs play a central role in postnatal CB development by being the main source of sonic hedgehog (SHH), which is required for proliferation of granule cell progenitors and Nestin-expressing progenitors that produce interneurons and astrocytes (*Corrales et al., 2006*; *Fleming et al., 2013*; *Lewis et al., 2004*). PCs also are key for CB function by integrating the inputs that converge on the cerebellar cortex (*Sillitoe and Joyner, 2007*). Hence, PC loss is linked to cerebellar motor behavior syndromes and has also been implicated in autism (*Fatemi et al., 2012*; *Tsai et al., 2012*; *Wang et al., 2014*). In this study we determined the regenerative potential of PCs in neonatal mice.

## Results and discussion

To ablate and track PCs, the diphtheria toxin receptor (DTR) and a lineage tracer, TdTomato (TdT), were expressed in a subpopulation of PCs using a transgenic approach ($Pcp2^{Cre/+}$; $R26^{LSL-DTR/LSL-TdT}$ or *PC-DTR* mice; LSL = lox stop-lox). We found that only $52.16 \pm 21.84\%$ of PCs (n = 5 mice), identified by expression of CALB1, expressed TdT and DTR at postnatal day (P) 1, and surprisingly the percentage and large variation remained similar at P5 and P30 (*Figure 1—figure supplement 1*). Strikingly, when DT was injected at P1 into *PC-DTR* pups (P1-*PC-DTR*), nearly all TdT+ PCs formed an ectopic layer below the PC layer (PCL) by 1 day post injection (dpi) (*Figure 1A–M*). The ectopic layer was absent by P8 (*Figure 1K*), and TdT+ cells in the ectopic layer were TUNEL positive starting at P3 with a peak at P5. These results show that almost all DTR-expressing TdT+ cells become misplaced, die and are cleared within 5–7 dpi of DT (*Figure 1N,O*).

Unexpectedly, although the number of CALB1+ PCs in the PCL of P1-*PC-DTR* mice was significantly reduced at P2 compared to non-injected controls (No DT), it was not significantly reduced at

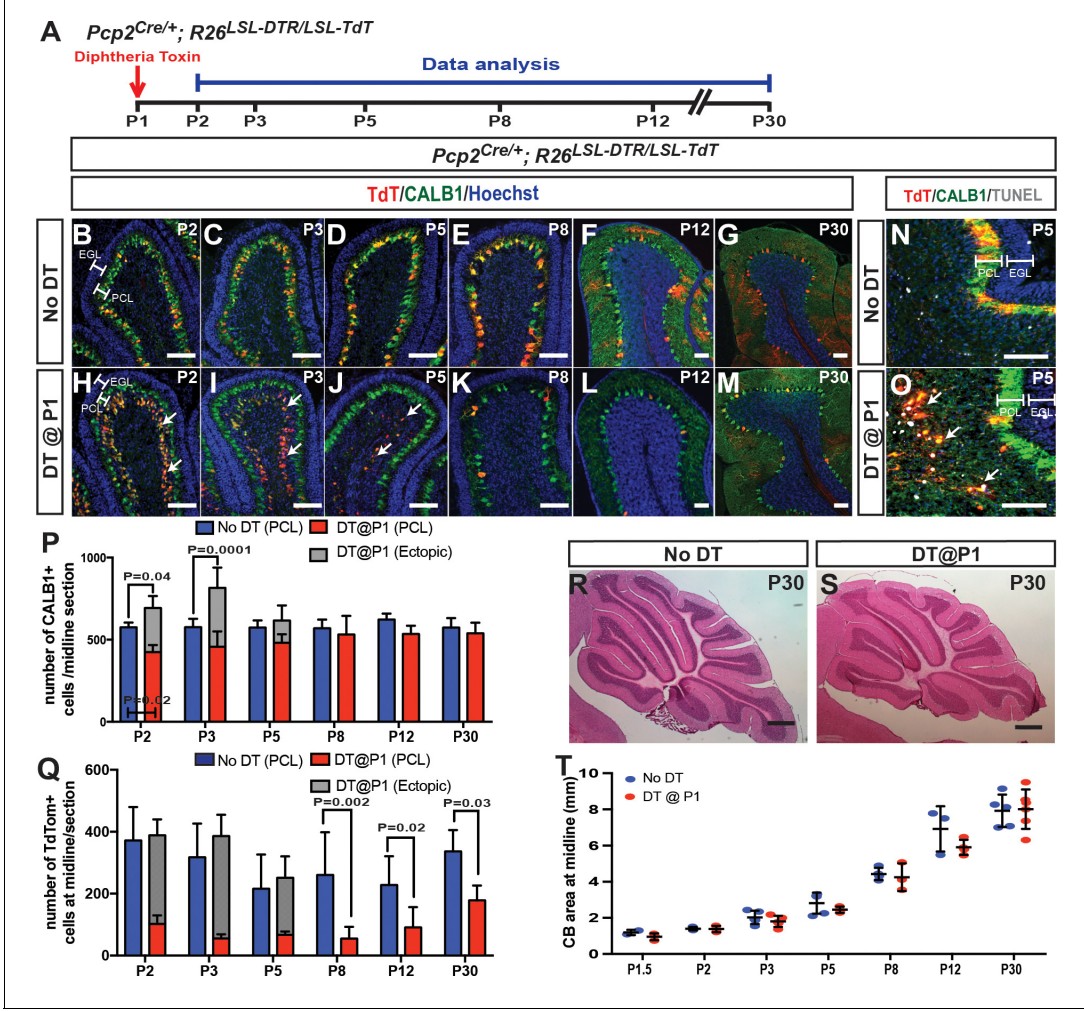

**Figure 1.** Ablation of PCs at P1 stimulates their replenishment and development of normal CB size and morphology. (**A**) The experimental plan. (**B–M**) IF analysis at the indicated ages for TdT and CALB1 in sagittal cerebellar sections of lobule IV-V in No DT (B-G) and P1-*PC-DTR* mice (H-M). (**N–O**) Analysis of apoptosis at P5 using TUNEL. (**P**) Quantification of CALB1+ cells per midline section in PCL (blue or red) and ectopic layer (grey) (PCL cells: Two-way ANOVA $F_{(5,54)}$=4.034, p=0.0035, and total number of PCs: Two-way ANOVA $F_{(5,27)}$=4.732, p=0.003, n ≥ 3 animals/condition). (**Q**) Quantification of TdT+ cells per section (PCL cells: Two-way ANOVA $F_{(5,48)}$=6.957, p=0.0001). Significant *post hoc* comparisons are shown. (**R–S**) H and E stained midline sagittal sections of cerebella at P30 of No DT (**R**) and P1-*PC-DTR* (**S**) mice. (**T**) Quantification of midline sagittal areas of cerebella shows no differences upon DT injection (p=0.89, n ≥ 3 for each age). Scale bars: (**B–O**) 200 μm, (**R–S**) 500 μm. (EGL: external granule layer, PCL: Purkinje cell layer).

DOI: https://doi.org/10.7554/eLife.39879.003

The following source data and figure supplements are available for figure 1:

**Source data 1.** Summary of the antibodies used in the study.

DOI: https://doi.org/10.7554/eLife.39879.007

**Source data 2.** Summary of the statistics performed.

DOI: https://doi.org/10.7554/eLife.39879.008

**Figure supplement 1.** DTR and TdT are co-expressed in ~50% of PCs in *PC-DTR* mice at P1, P5 and P30.

DOI: https://doi.org/10.7554/eLife.39879.004

**Figure supplement 2.** CB size and morphology appears normal following DT-mediated ablation of PCs at P1.

DOI: https://doi.org/10.7554/eLife.39879.005

**Figure supplement 3.** External granule cell layer thickness is not changed after DT-mediated killing of PCs at P1.

DOI: https://doi.org/10.7554/eLife.39879.006

P3 and later stages (*Figure 1P*). Furthermore, the total number of PCs (ectopic layer + PCL) was significantly greater in DT-injected cerebella than in No DT controls at P2 and P3, and the total number of PCs was down to normal levels at P5, overlapping with the time of clearance of the ectopic layer (*Figure 1P*). Although the number of TdT+ cells in the PCL increased between P8 and P30 in *P1-PC-DTR* brains, it remained significantly lower than in No DT controls at P30 (*Figure 1Q*). Given that there is no significant increase in the recombination induced by $Pcp2^{Cre}$ after birth in the control postnatal CB (*Figure 1Q*, *Figure 1—figure supplement 1F*), the percentage of TdT+ cells in *P1-PC-DTR* brains at P30 (~25–30%) matched the predicted percentage if 50% of the PCs were killed by DT and then ~50% of the regenerated PCs underwent recombination. Interestingly, and consistent with the rapid recovery of PC numbers in the PCL, no significant decrease in the sectional area of the CB was observed between P1.5 and P30 (*Figure 1R–T*, *Figure 1—figure supplement 2*). Furthermore, the thickness of the outer (proliferating) and inner (differentiating) external granule cell layers remained normal (*Figure 1—figure supplement 3*). In summary, we uncovered that the CB can rapidly recover (within 24–48 hr) from the loss of ~50% of PCs at P1, by producing new PCs and resuming normal growth.

In order to document the rapid production of new PCs after ablation, we tested whether PCs that had recently undergone cell division could be detected at P3. *P1-PC-DTR* mice were divided into four groups; each group receiving three injections of BrdU (2 hr apart) during 4–26 hr after DT-injection (*Figure 2A*). As predicted, BrdU+ PCs (FoxP2+ and CALB1+) were observed in the PCL of all groups (*Figure 2B*, *Figure 2—figure supplement 1*), with the greatest incorporation being between 10–20 hr after DT (*Figure 2C*, *Figure 2—figure supplement 1*). Importantly, in No DT mice no incorporation of BrdU was observed in PCs (*Figure 2—figure supplement 1*). Curiously, FOXP2 and BrdU showed non-overlapping subnuclear localization in the nuclei of PCs. In addition, when we analyzed brains of *P1-PC-DTR* mice at P30 that had received BrdU 10–14 hr after DT injection, we observed BrdU+ mature PCs with similar cell bodies and dendritic trees to their neighbors, showing that the newly generated PCs differentiate and survive to adulthood (*Figure 2—figure supplement 2*). Furthermore, a lack of BrdU incorporation in the ectopic layer at P3 shows that the labeling of PCs is not due to DNA damage induced by DT-mediated cell death (*Figure 2—figure supplement 1C*).

In order to further confirm that BrdU incorporation is not due to DNA damage following DT injection, we treated P1 pups either with 4Gy γ-irradiation or DT at P1 followed by three BrdU injections (2 hr apart) at 30 min or 10 hr after treatment, respectively. The brains were then analyzed 24 hr after the last BrdU injection. In the irradiated pups we observed extensive γ-H2AX foci, including in PCs, but BrdU incorporation was not detected in any PCs. In contrast, *P1-PC-DTR* mice injected with DT showed BrdU incorporation in PCs without any γ-H2AX foci (*Figure 2—figure supplement 3*). Thus, DNA damage does not account for the BrdU incorporation into PCs following ablation of ~50% of PCs at P1.

As a second means to specifically label dividing cells that give rise to new PCs, we intracranially injected GFP-expressing retrovirus into *P1-PC-DTR* pups and littermate controls 12 hr after DT injection, since retroviruses can only incorporate into the DNA of dividing cells and are widely used for clonal analysis of neural stem cells and progeny (*Figure 2D*)(*Cepko, 1988*; *Yu et al., 2009*). When the mice were analyzed at P21, we indeed observed rare GFP-labeled CALB1+ PCs in *P1-PC-DTR* animals near the site of injection, and not in No DT controls. As expected, GFP+ granule cells and Bergmann glia were observed in both the No DT and *P1-PC-DTR* mice (*Figure 2E–I*). These three sets of experiments thus reveal that a progenitor capable of proliferating produces the new PCs after ablation at P1.

Based on the rapid replenishment of PCs after ablation at P1, we hypothesized that a local progenitor in the PCL must be responsible for the response. The Nestin-expressing progenitors (NEPs) in the PCL were a candidate, as they display plasticity upon ablation of granule cell precursors in newborn mice (*Wojcinski et al., 2017*). Furthermore a putative rare Nestin+ cell in the adult CB was recently described as able to produce new neurons in response to exercise (*Ahlfeld et al., 2017*). However, when we tested the contribution of NEPs to PC regeneration using a *Nes-CFP* reporter allele that transiently maintains CFP protein after differentiation, no CFP+ cells were found to co-express FOXP2 or CALB1 at 12 hr and 2 days post DT injection in *P1-PC-DTR* mice and in No DT controls (*Figure 2—figure supplement 4*). Furthermore, inducible fate mapping of NEPs using *Nestin-FlpoER/+; R26^{FSF-TdT/+}* (FSF = frt stop-frt) mice showed no TdT+ PCs at P30 in *P1-PC-DTR* and

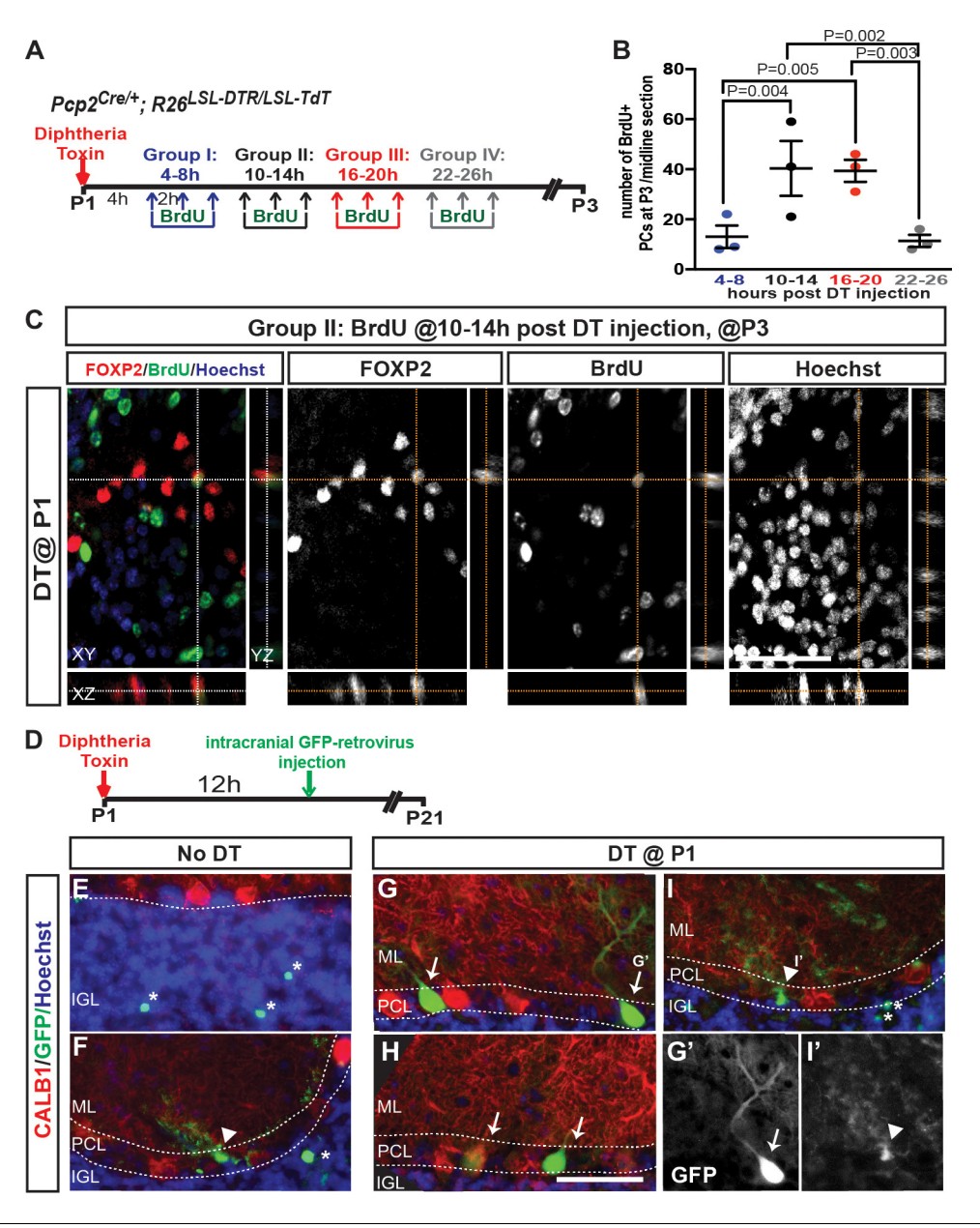

**Figure 2.** Progenitors proliferate within 24 hr of DT-injection at P1 in *PC-DTR* mice and produce new PCs. (**A**) The experimental plan. (**B**) Quantification of the number of BrdU+ PCs (CALB1+) at P3 in P1-*PC-DTR* mice (Two-way ANOVA $F_{(3,16)}$=6.163, p=0.006, n = 3 animals/condition). Significant *post hoc* comparisons are designated in the figure. (**C**) Representative images of BrdU injection performed at 10–14 hr post DT injection in P3 P1-*PC-DTR* CB. (n = 3 animals/condition). Orthogonal view from z-stack obtained by confocal microscopy demonstrates colocolization of BrdU and FOXP2 after PC depletion. (**D**) Experimental plan for retroviral labeling. P1-*PC-DTR* pups were injected with GFP expressing retrovirus to label proliferating cells and the brains were analyzed 3 weeks later. (**E–I**) IF analysis shows examples in two mice of each genotype of rare retrovirus labeled PCs (arrows) following regeneration (**G, H**) only in P1-*PC-DTR* mice. GFP+ GCs (asterisk) and Bergmann glia cells (arrow head) were observed in No DT (**E, F**) and *P1-PC-DTR* (**I**) mice (n = 6 mice/condition). G' and I' shows GFP-expressing PC and a Bergmann glia cell. Scale bars: 50 μm.

DOI: https://doi.org/10.7554/eLife.39879.009

The following figure supplements are available for figure 2:

**Figure supplement 1.** BrdU incorporation occurs within 24 hr of DT injection at P1 in *PC-DTR* mice and BrdU + PCs can be observed at P3.

*Figure 2 continued on next page*

*Figure 2 continued*

DOI: https://doi.org/10.7554/eLife.39879.010

**Figure supplement 2.** BrdU is detected in adult PCs of P1-*PC-DTR* animals that received BrdU 10–14 hr post DT injection.

DOI: https://doi.org/10.7554/eLife.39879.011

**Figure supplement 3.** DNA damage following irradiation or depletion of PCs in P1-*PC-DTR* mice does not result in incorporation of BrdU.

DOI: https://doi.org/10.7554/eLife.39879.012

**Figure supplement 4.** Nestin-expressing progenitors are not responsible for the recovery of PCs following DT-mediated ablation at P1.

DOI: https://doi.org/10.7554/eLife.39879.013

**Figure supplement 5.** Fate mapping confirms that Nestin-expressing progenitors are not responsible for the recovery of PCs following DT-mediated ablation at P1.

DOI: https://doi.org/10.7554/eLife.39879.014

No DT control mice given tamoxifen at P0 (*Figure 2—figure supplement 5*). These results suggest that a progenitor other than NEPs mediates regeneration following PC depletion.

We next examined whether a progenitor exists after birth that expresses early (FOXP2) but not late (CALB1) PC markers. Indeed, at P1 we identified CALB1 negative/low and FOXP2-expressing cells that could be immature PCs (named iPC for immature Purkinje cells; *Figure 3A–B*, *Figure 3—video 1*). Possibly accounting for the regeneration of PCs, iPCs were not labeled by *Pcp2^{Cre}* as they were TdT and DTR negative in No DT controls, thus they escape DT-mediated cell death (*Figure 3—figure supplement 1*). Temporal analyses revealed a decrease in the number of iPCs from P1 (74.33 ± 5.69/midline sagittal section) to P5 (28.66 ± 7.51/midline sagittal section, *Figure 3A*, *Figure 3—figure supplement 1*), indicating the progenitors are a transient population. Interestingly, the few iPCs present at P5 were specifically enriched in the central and nodular zones of the CB, which are developmentally delayed at P5 (*Legué et al., 2015*; *Sudarov and Joyner, 2007*) (*Figure 3A*).

In order to investigate the normal fate of iPCs, we tested whether there is an increase in the number of CALB1+ PCs from P1 to P30. In order to minimize variation across animals, we used the C57BL/6 inbred strain of mice and analyzed the entire half-vermis of each brain (every second section). As expected, a significant reduction in the number of iPCs was observed between P1 and P5, but in addition we detected a significant increase in the number of CALB1+ cells at P30 compared to P1 (*Figure 3C*). At P5 there was a trend towards an increase in the number of CALB1+ cells. There was also a trend towards a reduction in the total number of iPCs plus CALB1+ cells at P5 and P30 compared to P1 (*Figure 3C*), suggesting that some PCs may also undergo apoptosis during early postnatal development. These results provide evidence that iPCs are cells destined to become PCs, but normally undergo a delay in differentiation until the first week after birth.

We then asked whether the number of iPCs increases after DT treatment of P1-*PC-DTR* mice. Quantification of iPC numbers showed a significant increase 12 hr after DT injection in P1-*PC-DTR* mice (1.90 ± 0.05 fold, *Figure 3D*), correlating with the time window of highest BrdU incorporation after injury (*Figure 2B*). Interestingly, at P5 the number of iPCs was significantly lower in P1-*PC-DTR* animals than in No DT mice (*Figure 3D*), possibly reflecting an exhaustion of the progenitor population by production of new PCs. To further show that iPCs expand in number after their neighbors are killed, we used constitutive FLP-based fate mapping in *FoxP2^{Flpo/+}; R26^{FSF-TdT/+}* mice to transiently mark and follow PCs and iPCs. We found that all CALB1+ PCs and iPCs expressed TdT at P1 (*Figure 3—figure supplement 2*), and as predicted, an increase in transiently fate mapped TdT + iPCs was seen in P1-*PC-DTR* mice 12 hr after DT injection at P1 compared to No DT controls (1.86 ± 0.46–fold, n = 3, *Figure 3—figure supplement 3*). Thus, iPCs expand in number after damage to neighboring PCs.

To confirm that iPCs undergo proliferation upon PC depletion, we injected BrdU or EdU 10–14 hr after DT and collected cerebella 1 hr (~P1.5) later. Other than glial progenitors and microglia seen in No DT controls (*Figure 3—figure supplement 4*), all additional BrdU+ (or EdU+) cells in the PCL of P1-*PC-DTR* mice expressed FOXP2, and of these cells 45.5 ± 1.1% expressed CALB1 (*Figure 3E–N*, *Figure 3—figure supplements 5,6*). Furthermore, the total number of FOXP2+ cells in the PCL that

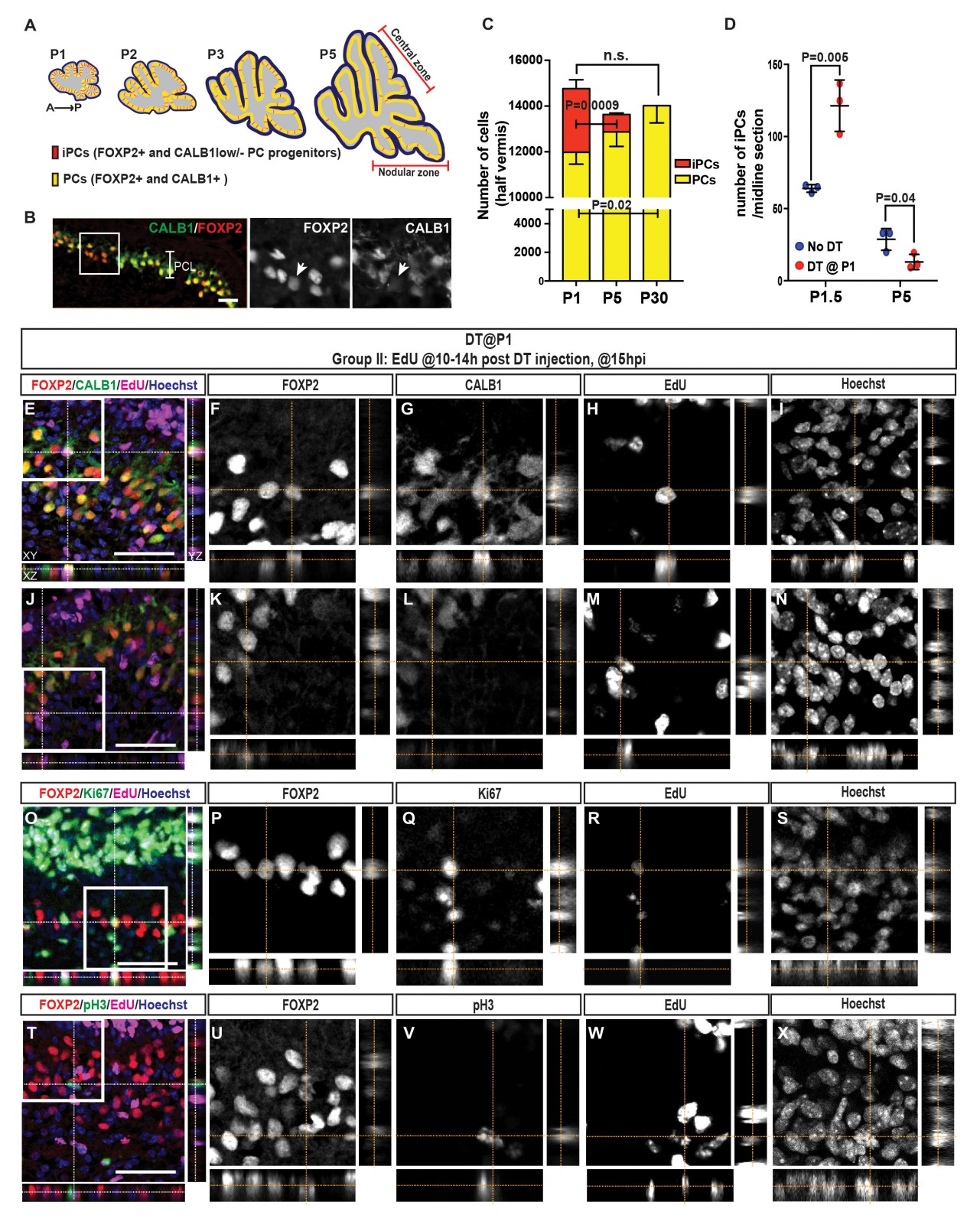

**Figure 3.** The number of iPCs diminishes with age and increases after ablation of PCs. (**A**) Schematic representation of the distribution of iPCs (red) in sagittal midline sections of P1-5 cerebella (yellow, FoxP2+ and CALB1+ PCs) (**B**) IF analysis of iPCs (FoxP2+ and CALB1-/low, arrow) at P1.5 in No DT mice. (**C**) Quantification of the numbers of iPCs and PCs at P1, P5 and P30 (CALB1+: One-way ANOVA $F_{(2.6)}$ = 6.883, p=0.028, iPCs: Student's t-test: p=0.0009, all cells: One-way ANOVA $F_{(2.6)}$ = 1.813, p=0.24, n = 3 animals/condition). Significant *post hoc* comparisons are shown. (**D**) Quantification of

*Figure 3 continued*

the numbers of iPCs at P1.5 (Two-tailed t-test, p=0.005, n = 3) and P5 (Two-tailed t-test, p=0.04, n = 3) in No DT and P1-*PC-DTR* mice. (E–N) Orthogonal projections of z-stack shows a EdU + PC (CALB1+, FOXP2+) (E–I) or iPC (CALB1-/low, FOXP2+) (J–N) at 15 hr post injection (hpi) in P1-*PC-DTR* mice (n = 3). (O–X) Orthogonal projections of z-stack shows EdU+ and FOXP2+ cells that either express the cell cycle markers KI67 (O–S) or pH3 (T–X) at 15 hr post injection (hpi) in P1-*PC-DTR* mice (n = 3). Scale bars: (B) 100 μm, (E, J, O, T) 50 μm.
DOI: https://doi.org/10.7554/eLife.39879.015

The following video and figure supplements are available for figure 3:

**Figure supplement 1.** iPCs are not labeled by *Pcp2*[Cre] and their numbers diminish with age.
DOI: https://doi.org/10.7554/eLife.39879.016

**Figure supplement 2.** *FoxP2-TdT* fate mapping marks iPCs in the PCL at P1 as well as rare cells outsidethe PCL.
DOI: https://doi.org/10.7554/eLife.39879.017

**Figure supplement 3.** The number of *FoxP2-TdT* transiently marked iPCs increases 12 hr after DT injection at P1.
DOI: https://doi.org/10.7554/eLife.39879.018

**Figure supplement 4.** Microglia and glial progenitors proliferate in both No DT and DT P1-*PC-DTR* mice.
DOI: https://doi.org/10.7554/eLife.39879.019

**Figure supplement 5.** IF analysis of PCs at P1.5 (15h post DT injection at P1) shows that FoxP2+ cells proliferate and there are more FoxP2+ CALB1-/low cells that incorporate BrdU than FOXP2+ CALB1+ high cells.
DOI: https://doi.org/10.7554/eLife.39879.020

**Figure supplement 6.** IF analysis of PCs at P1.5 (15h post DT injection at P1) shows that FOXP2+ cells proliferate (Ki67+ or pH3+) and BrdU+ FOXP2 + CALB1-/low cells can be observed at P1.5 but not at P3.
DOI: https://doi.org/10.7554/eLife.39879.021

**Figure supplement 7.** Analysis of P27 and Ki67 fluorescence intensity of iPCs and CALB1+ PCs in P1 wild type mice.
DOI: https://doi.org/10.7554/eLife.39879.022

**Figure 3—video 1.** Three-dimensional projection of a z-stack from the PCL of a P1 CB showing FoxP2+ CALB1 low/- iPCs.
DOI: https://doi.org/10.7554/eLife.39879.023

were acutely labeled with BrdU, was similar to the number of BrdU+ cells that became PCs (CALB1 +) at P3 (38.7 ± 9.1/section vs 40.3 ± 19.0/section, n = 3, *Figure 3—figure supplement 5*). In addition, FOXP2+ cells that were Ki67+ (*Figure 3O–S*, *Figure 3—figure supplement 6C*) or pH3+ (*Figure 3T–X*, *Figure 3—figure supplement 6D*) and EdU+ were detected at P1.5 in the PCL of P1-*PC-DTR* pups, confirming the presence of proliferative iPCs following PC ablation. In order to further study the cell-cycle state of iPCs in uninjured cerebella, we analyzed the expression levels of the cell cycle inhibitor P27[Kip1] (*Watanabe et al., 1998*) and KI67 in iPCs compared to CALB1 high FOXP2 + PCs at P1. Fluorescent intensity analyses revealed that PCs have higher P27 and lower KI67 expression levels compared to iPCs (*Figure 3—figure supplement 7*). Collectively, our data argues that the recovery of PCs in P1-*PC-DTR* mice is mediated by a previously unrecognized and age-dependent progenitor population (iPCs) that normally transitions to a CALB1+ PC, but in response to loss of PCs proliferates and differentiates to replace the lost cells.

Given that the population of iPCs is greatly reduced by P5 (*Figure 3C*), PCs should not be efficiently replaced if ablated at P5, when similar to at P1 *Pcp2*[Cre] induces recombination (expression of DTR) in 40.5 ± 21.5% of CALB1+ PCs (*Figure 1—figure supplement 1*). As predicted, when DT was injected at P5 (P5-*PC-DTR* mice) (*Figure 4—figure supplement 1A*), the number of PCs was significantly reduced at P12 compared to No DT controls (*Figure 4—figure supplement 1B–I,R*). TdT + PCs were TUNEL+ by P8 (*Figure 4—figure supplement 1J–K*) and the majority of TdT+ cells were cleared from the PCL by P12 (*Figure 4—figure supplement 1G,P and S*). Furthermore, in P5-*PC-DTR* mice at P8 and P12 the dendrites and cell bodies of the PCs were poorly organized compared to in controls (*Figure 4—figure supplement 1B–G and L–P*) and at P30 the cell bodies of some PCs were misplaced into the molecular layer (*Figure 4—figure supplement 1N–R*). Importantly, the reduction in PC numbers observed at P12 was maintained at P30 (*Figure 4—figure supplement 1R*), such that the number of PCs was reduced by 32.4 ± 6.5%. In summary, there is little replenishment of PCs when they are ablated at P5 (*Figure 4A*).

We next tested whether the rare iPCs at P5 (*Figure 3A*) can still proliferate upon PC depletion. In contrast to P1-*PC-DTR* mice, very few iPCs/PCs were BrdU+ in P5-*PC-DTR* cerebella injected with BrdU at 10–14 hr post DT-injection at both 1 hr (5.55 ± 0.51/ midline sagittal section, n = 3) and 1.5 days (6.22 ± 1.07/ midline sagittal section, n = 3, *Figure 4B*) post BrdU-injection. The few BrdU

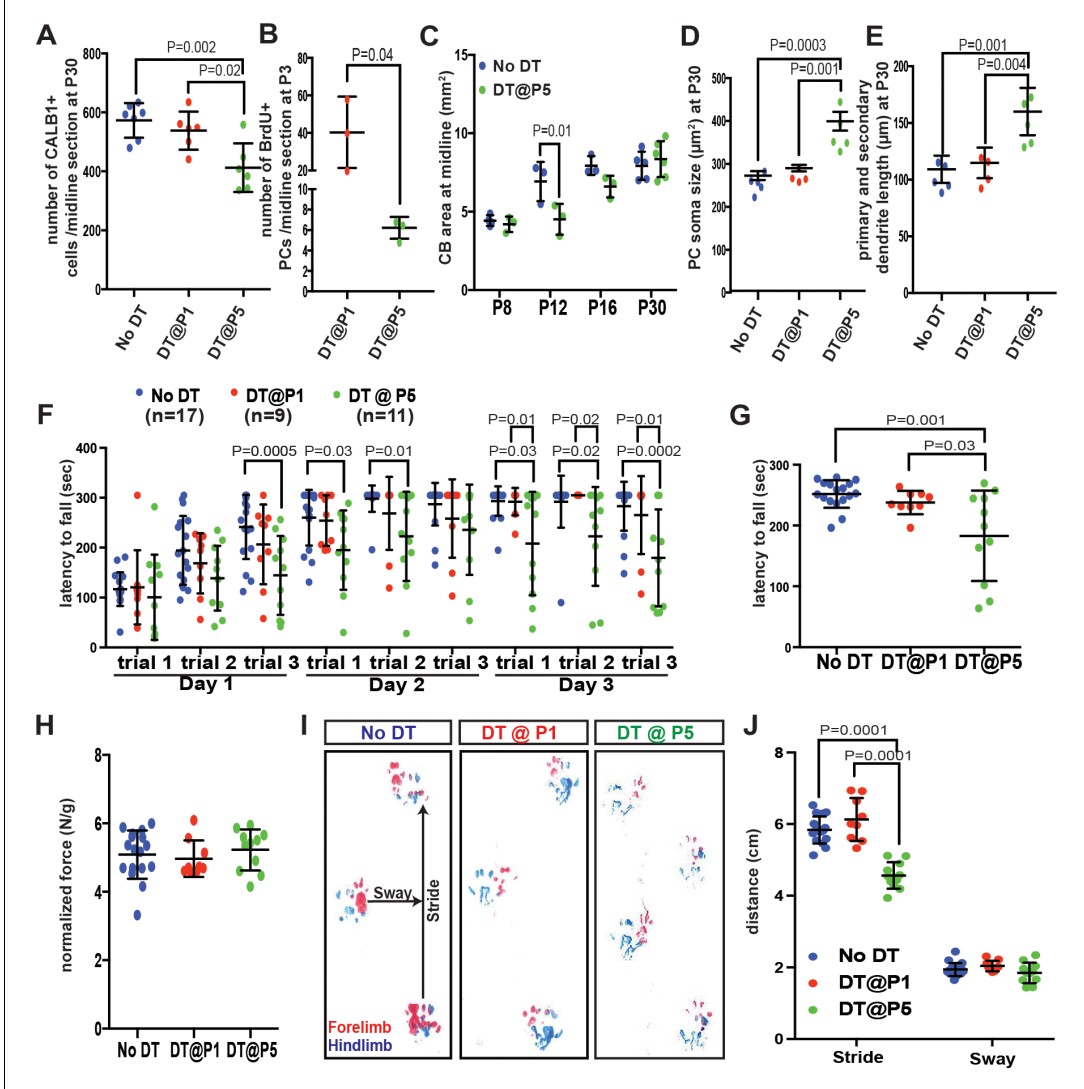

**Figure 4.** Despite the recovery of CB size, PCs are poorly replenished and motor behavior deficits develop when PCs are killed at P5 but not at P1.
(**A**) Number of CALB1+ cells at P30 (One-way ANOVA, $F_{(2,16)}$=9.464, p=0.002, n ≥ 6). (**B**) Number of BrdU+ PCs 2 days post DT-injection in P1- or P5-*PC-DTR* mice (Two-tailed t-test, p=0.04). (**C**). Quantification of CB area in midline sagittal sections demonstrates that CB size is smaller at P12 in P5-*PC-DTR* mice but not later (Two-way ANOVA, $F_{(1,22)}$=7.045, p=0.01, n ≥ 3). (**D–E**) PC soma size (D, One-way ANOVA, $F_{(2,11)}$ = 20.56, p=0.0002, n ≥ 4) and primary and secondary dendrite lengths (E, One-way ANOVA, $F_{(2,11)}$=14.54, p=0.0008, n ≥ 4) at P30 were increased in P5-*PC-DTR* animals compared to No DT and P1-*PC-DTR* animals. (**F–G**) Latency to fall from rotarod at each trial (F, Two-way ANOVA, $F_{(2,34)}$=8.37, p=0.001, n ≥ 9) and cumulative analysis (G, One-way ANOVA, $F_{(2,34)}$=11.12, p=0.0002, n ≥ 9, No DT vs. DT@P1: p=0.83) for P30 P5-*PC-DTR* animals compared to No DT and P1-*PC-DTR* animals. (**H**) Analysis of grip strength showed no change in P1 (n = 9, vs No DT: p=0.89) and P5 (n = 11, vs. No DT: p=0.84, vs. DT@P1: p=0.64) DT-injected mice compared to controls (No DT, n = 17). (**I–J**) Representative images (I) and quantification (J) of footprint analysis performed on P1- (vs. No DT: stride: p=0.10 and sway: p=0.90) and P5-*PC-DTR* mice and controls (Two-way ANOVA, $F_{(2,133)}$=73.45, p=0.0001, n ≥ 9). Significant *post hoc* comparisons are shown.

DOI: https://doi.org/10.7554/eLife.39879.024

The following figure supplements are available for figure 4:

**Figure supplement 1.** PC numbers are reduced upon PC ablation at P5 in *PC-DTR* mice.
DOI: https://doi.org/10.7554/eLife.39879.025

**Figure supplement 2.** Distribution of BrdU+ PCs in P5-*PC-DTR* mice at 15 hr and 2 days post injection of DT.
DOI: https://doi.org/10.7554/eLife.39879.026

**Figure supplement 3.** Transient decrease in CB size and altered PC morphology after ablation of PCs at P5.
DOI: https://doi.org/10.7554/eLife.39879.027

**Figure supplement 4.** Transient decrease in external granule cell layer thickness after DT injection at P5.

*Figure 4 continued on next page*

*Figure 4 continued*

DOI: https://doi.org/10.7554/eLife.39879.028

**Figure supplement 5.** Graphical summary of the findings.

DOI: https://doi.org/10.7554/eLife.39879.029

+ iPCs/PCs present were concentrated in the central and the nodular zones that are enriched for iPCs at P5 (*Figure 4—figure supplement 2*). Interestingly, compared to P1-*PC-DTR* mice in which 52.29 ± 0.09% (n = 3) of iPCs incorporated BrdU, only 20.55 ± 0.07% (n = 3) incorporated BrdU in P5-*PC-DTR* animals. Overall, these results demonstrate that replenishment of PCs is not efficient at P5 because with age, iPCs both diminish in number and in their ability to proliferate in response to PC depletion.

We next examined whether the depletion of PCs in P5-*PC-DTR* mice had an effect on CB development. Indeed, the area of CB sections was significantly reduced at P12 but not P8 (*Figure 4C*, *Figure 4—figure supplement 3* although the thickness of the external granule cell layer was significantly reduced in P5-*PC-DTR* mice at P8. By P12 the thickness of the external granule cell layer was similar in PC-ablated mice and controls (*Figure 4—figure supplement 4A–E*). Surprisingly, despite the lack of recovery of PC numbers we found that the area of the CB was normal at P16 and P30 (*Figure 4C*, *Figure 4—figure supplement 3A–I*). As a consequence, there was a reduction in PC density compared to No DT or to P1-*PC-DTR* mice (*Figure 4—figure supplement 3J*, *Figure 4—figure supplement 2N,Q*). The density of granule cells also was lower compared to No DT and P1-*PC-DTR* P30 mice (*Figure 4—figure supplement 4F*). Interestingly, PCs in P5-*PC-DTR* mice had a larger soma (*Figure 4D*) and longer primary and secondary dendrites (*Figure 4E*) compared to No DT or P1-*PC-DTR* mice, a cellular phenotype observed in some mouse mutants with PC loss (*Castagna et al., 2016*). Furthermore, compared to controls, the percentage of PCs present at P30 in P5-*PC-DTR* animals compared to No DT controls (~66%) did not match the percentage of granule cells that were produced (~81% of No DT controls). Thus, the ratio of PCs to granule cells is disrupted in P5-*PC-DTR* animals because granule cells are over-produced. These results reveal that independent of iPCs being stimulated to produce new PCs following their ablation, there are mechanisms of cell and organ size regulation that ensure recovery of CB size.

Finally, given that the circuitry (proportions of neurons) is disrupted in P5-*PC-DTR* mice and not P1-*PC-DTR* mice but CB size is normal in both, we tested whether either mutant has normal motor function at P30. Interestingly, P1-*PC-DTR* animals had no significant changes in their motor function compared to controls (*Figure 4F–J*), whereas P5-*PC-DTR* mice showed a significant reduction in their latency to fall from the rotarod and had a shorter stride compared to both No DT and P1-*PC-DTR* mice (*Figure 4F–G and I–J*) but no change in grip strength (*Figure 4H*). These results demonstrate that P5-*PC-DTR* mice, but not P1-*PC-DTR* mice, have motor behavior deficits. Thus, rapid production of new PCs by iPCs enables establishment of functional circuits following depletion of PCs at P1. Furthermore, achieving correct cell numbers and/or proportions appears to be more important than maintaining CB size for functional recovery after injury in P5-*PC-DTR* mice.

In summary, we discovered a new regenerative process in the developing CB involving a previously unidentified and normally dormant and immature PC progenitor (iPC) that is able to expand and produce additional PCs, likely to buffer against early postnatal loss of these postmitotic neurons due to injury. Proliferation of iPCs is stimulated by ablation of PCs at P1 and importantly the response is rapid (10–48 hr), ensuring other components of the developing CB that are dependent on PCs for their proliferation or differentiation are not compromised. However, iPCs decrease in number and their capacity to proliferate during the first postnatal week, and consequently PCs are poorly replenished when ablated at P5. The cerebella of P5-*PC-DTR* mice nevertheless try to adapt by attaining near normal dimensions through a mechanism that includes increasing cell size (*Figure 4—figure supplement 5*). The CB therefore has multiple mechanisms for regulating organ size following perinatal injury that depend on the precise stage of development. Furthermore, the motor deficits seen in P5-*PC-DTR* mice highlight the importance of maintaining the correct number of PCs and relative neuron proportions during development, not just organ size.

One possible reason for why iPCs differentiate into PCs after P1 and lose their ability to proliferate is that a critical component of the microenvironment that supports iPCs is diminished soon after birth, perhaps as a consequence of a developmental clock that the cells in the microenvironment

follow. A second possibility is that the differentiation of iPCs is dictated by the timing of the establishment of the cerebellar circuitry. We speculate that efficient regeneration is possible at P1 because PCs still have an immature morphology and integration into the cerebellar circuitry, whereas at later stages the parallel fibers (axons of granule cells) synapse with PCs and climbing fibers (axons of the inferior olive neurons) refine their synapses and both cells promote PC maturation (*Good et al., 2017*; *Hoxha et al., 2017*). Thus, maturation and integration of a newly generated PC into the cerebellar circuitry might not be efficient or possible after P5. By extrapolation, the replenishment process has evolved such that developmental plasticity is tightly correlated with age-dependent maturation of the neural circuit.

An additional cellular process to consider for the age-dependency of regeneration is the ability of neurons to enter back into the cell-cycle. Most differentiated neurons, including PCs, when forced to proliferate undergo apoptosis (*Feddersen et al., 1992*). However, previous reports have shown that following experimental manipulation or neurodegeneration, ectopic proliferation of adult retinal and pyramidal neurons can occur (*Ajioka et al., 2007*; *Sage et al., 2005*; *Skapek et al., 2001*; *Yang et al., 2001*). Our data indicate that iPCs, which lack the mature PC marker CALB1 and express the immature marker FOXP2, show low expression of P27 and weak but higher expression of KI67 compared to CALB1+ PCs, suggesting that their cell-cycle exit may be incomplete. CALB1+ PCs, likely ones that recently began making CALB1 protein, also appear to be able to re-enter the cell-cyle. However, the increase in the number of iPCs ~12 hr after DT administration suggests that the main regenerative response is achieved by the proliferation of iPCs.

The regenerative processes previously described in neuronal tissues involve adaptive reprograming of cells that are either actively proliferating or retain proliferative capacity and also have cell fate plasticity (*Benner et al., 2013*; *Buffo et al., 2008*; *Jinnou et al., 2018*; *Lin et al., 2017*; *Llorens-Bobadilla et al., 2015*; *López-Juárez et al., 2013*; *Martí-Fàbregas et al., 2010*; *Robel et al., 2011*; *Samanta et al., 2015*; *Sirko et al., 2013*; *Wojcinski et al., 2017*). Here we identified a distinct regenerative process that involves a local and normally dormant or immature progenitor. Unlike NEPs of the CB or astrocytes and neural stem cells in the forebrain that produce neurons upon injury, iPCs do not require reprograming and cell fate plasticity as our data indicates that they normally produce additional CALB1+ PCs after birth. Thus, iPCs maintain their lineage decision, but proliferate and then mature upon injury. An important question raised by our study is whether regeneration of postmitotic neurons by age-dependent progenitors is unique to the CB where protracted development might provide a conducive milieu, or whether all brain regions retain similar progenitors for a particular time window after each neuron subtype is generated. Furthermore, understanding the mechanisms of PC regeneration in newborn mice could provide insights into how regeneration in the adult brain can be enabled.

## Materials and methods

### Animals

All the experiments were performed according to protocols approved by the Memorial Sloan Kettering Cancer Center's Institutional Animal Care and Use Committee (IACUC). Animals were given access to food and water *ad libitum* and were housed on a 12 hr light/dark cycle.

The following mouse lines were used for these experiments: $Pcp2^{Cre}$ (*Zhang et al., 2004*), Nestin-CFP (*Mignone et al., 2004*; *Wojcinski et al., 2017*), Nestin-FlpoER (*Wojcinski et al., 2017*), $FoxP2^{Flpo}$ (*Bikoff et al., 2016*), $Rosa26^{LSL-DTR}$ (Stock no: 007900, The Jackson Laboratories) (*Buch et al., 2005*), $Rosa26^{LSL-TdT}$ (*ai14*, Stock no: 007909, The Jackson Laboratories) (*Madisen et al., 2010*), $Rosa26^{FRT-STOP-FRT-TdT}$ derived from Ai65 (Stock no: 021875, The Jackson Laboratories)(*Madisen et al., 2015*), C57BL/6J (Stock no: 00664, The Jackson Laboratories). Both sexes were used for analyses and no randomization was used. Exclusion criteria for experimental data points were sickness or death of animals during the testing period. No randomization was used and masking was used only for the behavior studies where the experimenter was blind to the genotypes.

Diphtheria toxin (30 µg/g of mouse; List Biological Laboratories, Inc.) was injected subcutaneously either at postnatal day (P) one or P5 and the brains were collected at various ages (*Figure 1a* and *Figure 4—figure supplement 1*). Mice not given DT (No DT mice) were $Pcp2^{Cre/+}$; $R26^{DTR/LSL-TdT}$

littermates and injected with the same volume of vehicle (PBS). BrdU or EdU (50 µg/g of mouse; Sigma) was injected subcutaneously.

## Tissue preparation and histological analysis

For P5 and younger animals, brains were dissected and fixed in 4% paraformaldehyde (PFA) for 24–48 hr (h) at 4°C. Animals older than P5 were anesthetized using intraperitoneal injection of a Ketamine (100 mg/kg) and Xylaxine (10 mg/kg) cocktail. Once full anesthesia was achieved, animals were systemically perfused with ice-cold PBS, followed by 4% PFA. Brains were dissected and post-fixed in 4% PFA for 24–48 hr. Fixed brains were allowed to sink in 30% Sucrose in PBS solution and then embedded in OCT (Tissue-Tek) for cryosectioning. 14 µm-thick cryosections were obtained using a Leica cryostat (CM3050S) and mounted on glass slides. Frozen sections were stored at −20°C for future analysis. In order to generate the 3D renderings in *Figure 3—video 1* 60 µm-thick cryosections were obtained and staining was performed on free floating sections. Haematoxylin and Eosin (H and E) staining was performed to assess cerebellar cytoarchitecture and measure area (size).

For immunofluorescent (IF) analysis, slides were allowed to warm to room temperature (RT). After washing once with PBS, slides were blocked using 5% Bovine Serum Albumin (BSA, Sigma) in PBS-T (PBS with 0.1% Triton-X) for 1 hr at RT. Slides were then incubated overnight at 4°C with primary antibodies diluted in blocking buffer. *Figure 1—source data 1.* summarizes the primary antibodies used in this study. Upon primary antibody incubation, slides were washed with PBS-T (3 × 5 min), incubated with specific AlexaFluor-conjugated secondary antibodies (1:500 in blocking buffer, Invitrogen) for 1 hr at RT and then washed again with PBS-T (3 × 5 min). Counterstaining was performed using Hoechst 33258 (Invitrogen) and the slides were mounted with Fluoro-Gel mounting media (Electron Microscopy Sciences). EdU was detected using a commercial kit following the manufacturer's recommendations (Invitrogen Cat no: C10340).

## Retrovirus injection

The super folding (sf)-GFP-expressing VSVG-pseudotyped gamma-retrovirus (Moloney murine leukemia virus) was made in HEK293T (ATCC #CRL-11268) cells using the pCMV-vsvg and pCMV-gp packaging plasmids and pUX-sf-GFP retrovirus vector plasmid (cloned by inserting sf-GFP into the BglII and NotI sites of the pUX plasmid (*Gu et al., 2011*) as previously described (*Yu et al., 2009*; *Zhao et al., 2006*). 10–12 hr after DT injection, P1 P1-*PC-DTR* pups were anesthetized by hypothermia. 3 µL of (sf)-GFP-expressing retrovirus particles (>$10^9$ Tu/mL) were injected intracranially into P1 vermal cerebella using a stereotactic apparatus. On average 12–15 sections were analyzed that were ~50 µm apart around the injection site. 7–9 retroviral-labeled PCs per mouse were detected only in the P1-*PC-DTR* brains (n = 6/ condition)

## Irradiation

An X-RAD 225Cx (Precision X-ray) microirradiator in the Small Animal Imaging Core Facility at Memorial Sloan Kettering Cancer Center was used to provide a single dose of 4 Gy irradiation, as previously described (*Wojcinski et al., 2017*), to P1 pups anesthetized by hypothermia. The CB was targeted using a collimator with 5 mm diameter.

## Image acquisition and analysis

Images were collected either with a DM6000 Leica microscope or Zeiss LSM 880 confocal microscope and processed using ImageJ Software (NIH).

For each quantification, three midline parasagittal sections/brain were analyzed and data was averaged. Cells were counted using the Cell Counter plugin for ImageJ (NIH). Analyses of the numbers of PCs and iPCs were performed by counting all of the PCs on a midline parasagittal section. CB area was calculated by defining a region of interest by outlining the perimeter of the outer edges of the CB, using ImageJ. EGL thickness was calculated by dividing the area of the EGL by the length of the EGL in midline sections. IGL density was calculated by counting the number of nuclei in three 40x fields from lobule eight in three midline sections and by dividing the number by the area of the region counted.

In order to reduce variation and address the fate of iPCs, we used P1-30 inbred mice (C57BL/6J) and analyzed half of the vermis. Analysis of the number of iPCs and PCs was performed on every

other section from 14 µm-thick sections to avoid counting the same cells twice due to their larger soma size. On average 25–28 sections were counted per brain.

Intensity measurements for P27 and KI67 expression in iPCs compared to CALB1+ PCs at P1 were performed using ImageJ. iPC or PC nuclei were defined as the region of interest and the marker fluorescence intensity and the nuclear area were measured and reported as corrected total cell fluorescence (CTCF)/nuclear area. (CTCF = Integrated Density – (Nuclear area X mean fluorescence of background readings).

PC soma size and dendrite length were calculated using randomly distributed TdT+ PCs from three midline sections (>20 cells/section). Soma area was calculated by outlining the perimeter of the outer edges of each cell. Cells that showed primary dendrites were used for this analysis to ensure that the region where the maximum soma area observed was used for the analyses. For dendrite length quantifications, primary and secondary dendrite length was measured and summed and PCs around the base of fissures were omitted.

## Behavioral testing

5 week old animals (No DT: n = 17, DT@P1: n = 9 and DT@P5: n = 11) were used to assess differences in motor behavior. The same sets of mice were used for all three tests described below.

### Rotarod

An accelerating rotarod (47650; Ugo Basile) was used for these experiments. Animals were put on the rod, and allowed to run till the speed reached to 5 rpm. Then the rod was accelerated from 5 to 40 rpm over the course of 300 s. Recording was stopped at 305 s. Time of fall was recorded for each animal. Analysis was performed three times a day on three consecutive days. Between each trial, animals were allowed to rest for 10 min in their home cage.

### Grip strength

To test whether any effects observed in the rotarod test were due to muscle weakness, grip strength measurements were performed using a force gauge (1027SM Grip Strength Meter with Single Sensor, Columbus Instruments). Animals were allowed to hold a horizontal grip while being gently pulled away by holding the base of their tail. Measurements were performed 5 times with 5 min resting periods in between. Force amount was recorded. Data was normalized to mouse's weight and represented in (Force/gram).

### Footprinting analysis

Forefeet and hindfeet were painted with red and blue nontoxic acrylic paint (Crayola), respectively. Animals were allowed to walk on a strip of paper laid along the floor of a 50 cm long, 10 cm wide custom-made Plexiglas tunnel with a dark box at the far end. Three runs/mouse were performed and the distances between the markings were measured.

## Statistical analysis

Prism (GraphPad) was used for all statistical analysis. Statistical comparisons used in this study were Student's two-tailed t-test; One-way and Two-way analysis of variance (ANOVA), followed by post hoc analysis with Tukey's test for multiple comparisons. Relevant F-statistics and p values are stated in the figure legends and the p values of the relevant post hoc multiple comparisons are shown in the figures. Summary of all the statistical analysis performed can be found in *Figure 1—source data 2*. The statistical significance cutoff was set at $p < 0.05$. Population statistics were represented as mean ± standard deviation (SD) of the mean. No statistical methods were used to predetermine the sample size, but our sample sizes are similar to those generally employed in the field. $n \geq 3$ mice were used for each experiment and the numbers for each experiment are stated in the figure legends.

## Acknowledgements

We thank past and present members of the Joyner laboratory for discussions and technical help. We thank T Jessell and Jay Bikoff for providing the *FoxP2^Flpo* line, P Faust for sending us the *Pcp2^Cre*

line and S Shi for providing the GFP-retrovirus. We are grateful to M E Hatten, S Shi, R Sillitoe, A Rosello-Diez and D G Placantonakis for comments on the manuscript. This work was supported by grants from the NIH to ALJ (R01NS092096 and R37MH085726) and a National Cancer Institute Cancer Center Support Grant [P30 CA008748-48].

## Additional information

### Funding

| Funder | Grant reference number | Author |
|---|---|---|
| National Institute of Neurological Disorders and Stroke | R01NS092096 | Alexandra L Joyner |
| National Cancer Institute | P30 CA008748-48 | Alexandra L Joyner |
| National Institute of Mental Health | R37MH085726 | Alexandra L Joyner |

The funders had no role in study design, data collection and interpretation, or the decision to submit the work for publication.

### Author contributions

N Sumru Bayin, Alexandre Wojcinski, Alexandra L Joyner, Conceptualization, Resources, Data curation, Formal analysis, Supervision, Funding acquisition, Writing—original draft, Writing—review and editing; Aurelien Mourton, Conceptualization, Data curation; Hiromitsu Saito, Noboru Suzuki, Resources

### Author ORCIDs

N Sumru Bayin http://orcid.org/0000-0003-4371-855X
Alexandra L Joyner http://orcid.org/0000-0001-7090-9605

### Ethics

Animal experimentation: All the experiments were performed according to protocols (#07-01-001) approved by the Memorial Sloan Kettering Cancer Center's Institutional Animal Care and Use Committee (IACUC). Animals were given access to food and water ad libitum and were housed on a 12-hour light/dark cycle.

### Decision letter and Author response

Decision letter https://doi.org/10.7554/eLife.39879.033
Author response https://doi.org/10.7554/eLife.39879.034

## Additional files

### Supplementary files

• Transparent reporting form
DOI: https://doi.org/10.7554/eLife.39879.030

### Data availability

All data generated or analysed during this study are included in the manuscript and supporting files.

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
