## [Decision Letter]

Thank you for submitting your work entitled "Age-dependent dormant resident progenitors are stimulated by injury to regenerate Purkinje neurons" for consideration by *eLife*. Your article has been reviewed by three peer reviewers, including Liqun Luo as the Reviewing Editor and Reviewer #1, and the evaluation has been overseen by a Senior Editor.

As you can see from the reviews, while finding the manuscript of potential interest, all three reviewers have reservations about the major conclusion. Should you be able to address these critiques with new experiments, we would be happy to consider a new manuscript in the future.

*Reviewer #1:*

Following their recent fascinating discovery on developmental plasticity of neurogenesis of cerebellar granule cells (Wojcinski et al., 2017), Joyner and colleagues here examine the developmental plasticity of cerebellar Purkinje cells (PCs). After application of diphtheria toxin (DT) to mice that express DTR using Pcp2-Cre at P1 (which is expressed in about half of the PCs at that time), they found that these DTR-expressing PCs were indeed killed by ~P5. However, PC number recovered nevertheless. They ascribe this recovery to the proliferation of a latent progenitor they termed FEPs because they express the early PC marker FOXP2 but not the mature PC marker Calbidin. They found that FEPs were present throughout the P1 cerebellum at substantial number, but diminished in number at P5. Accordingly, ablating Pcp2-Cre+ PCs at P1 resulted in a much less significant recovery of PC numbers; the residual PCs nevertheless increased in soma size and dendritic length. Mice with P1 but not P5 PC ablation recovered normal motor function by a few standard assays.

Overall the paper is of high technical quality and findings are interesting. However, I have one major reservation in their interpretation that the proliferating cells that produce PCs represent "a previously unidentified and normally dormant progenitor population." This proposal raises a number of questions: What is the normal function of these FEPs (they surely cannot just prepare for DT/DTR-induced cell killing)? What is their fate if no injury occurs? If we do not know their normal fate, how can we claim that they are "unipotent"? One alternative (and simpler) interpretation is that these FEPs are immature PCs that somehow can go back to cell cycle if their neighboring PCs are lost, and this ability diminishes as development proceeds (from P1 to P5). Indeed there is precedent in other neuronal types, such as horizontal cells in the retina, which can undergo cell division after fully differentiated in adults after loss of certain cell cycle inhibitor/tumor suppressor genes (Ajioka et al., 2007). It is possible that transient down-regulation of these cell cycle inhibitors in immature PCs, triggered by environmental signals due to loss of neighboring PCs, could cause immature PCs to re-enter the cell cycle. To me, this alternative developmental plasticity is no less interesting.

I have a few other technical critiques:

1) It seems unusual that there are fewer tdT+ neurons in the Pcp2-Cre;Ai14;LSLDTR mouse following DT injection (see Figure 1J-M and compare Figure 1P and 1Q). Is Pcp2-Cre, which should be expressed in all normal PCs after P7, no longer active in these "new" PC replacement cells? If so, how do we know these are real and/or functioning PCs?

2) In the P5 killing experiment, presumably by then Pcp2-Cre is expressed in a larger fraction of PCs compared with P1 (it would be useful to quantify this as done in P1 mice). Can this contribute to the difference between P1 and P5 ablation?

3) It appears that the authors deduce PC soma size from 14 µm sections. Given PC soma size is larger than 14 µm, this kind of analysis would produce a large number of half-cut, incomplete PC somal images, which might skew the conclusion?

*Reviewer #2:*

This manuscript describes regeneration of cerebellar Purkinje cells following diphtheria toxin injection at P1 but not at P5 and suggests that this regeneration is through cell division of a "previously unidentified progenitor population". Based on the data shown and previous studies of PC cell number, other possibilities exist to explain the findings. I am not convinced that the paper demonstrates generation of new Purkinje cells.

The specific comments are:

1) There is wide variability in the number of cells expressing Td-tomato/diphtheria toxin receptor at P1 (+/- 22%) which makes it difficult to know how many cells are dying. No counts of the number of cells expressing Td-tomato/diphtheria toxin are given for P5 mice. Presumably the percent of cells would be higher as Pcp2 is not fully expressed in all PCs at P1.

2) The representative image of DT treated PCs at P30 (Figure 1M) clearly shows a large number of missing PCs which would normally form a continuous monolayer, suggesting that the number of PCs did not recover from cell death. Is this a representative image?

3) Throughout the figures, co-labeling of PC markers with proliferation markers is not convincing. Images showing definitive co-localization should be provided.

4) In general, the quality of the images is not high and very few co-labeled cells are shown. It is not clear that some of the very few BrdU^+^ cells shown are not Bergmann glial cells. High quality images of BLBP stained BGs would be more convincing than the low power images provided. In general, higher magnification and resolution images would be potentially more convincing than the data shown.

5) Other studies have shown BrdU labeling of PCs in the adult, suggesting that BrdU or Thy incorporation assays detect DNA replication as well as cell division. Again, the co-labeling experiments with Ki67 and BrdU are not convincing. H3 would have been stronger evidence for cell division.

6) The authors suggest that Calb1+ PCs can develop within one day after a putative cell division. How do they imagine such cells would differentiate, i.e. extend dendrites into the field of parallel fibers? There are no images of the newly generated PCs that convince me that they are immature PCs.

7) There is literature suggesting that PCs undergo programmed cell death. While no direct studies have shown this, Bcl overexpression mice and Bax knockout mice both have ~30% more PCs. The authors should discuss these findings and how it might affect their results. If an immature (CALB1-) pool of PCs persists until P5, this would provide an alternative view to the idea that new cells are generated from "unidentified precursors".

*Reviewer #3:*

In this manuscript, Baylin et al. examined injury induced regeneration pf Purkinjie cells in neonates and proposed a model that upon elimination of Purkinje cells, a previously unknown quiescent progenitor population can be rapidly activated to replenish the loss cells (within 24hrs), resulting in normal cerebellum development. Furthermore, this progenitor population lost its capacity around P5. While the topic is interesting, the result is not convincing. High doses of BrdU is known to label DNA repair, especially in the context of massive injury and the authors mostly used protein markers for cell type identification. The process occurs so quickly (within 24 hrs) and in the fetal tissue. The authors should be able to use slice imaging to observe the death and birth of Purkinje cells directly, to illuminate any doubt of their model. In addition, the authors can use onco-retroviruses for lineage tracing to label new neurons and observe distinct Purkinje cell morphology.

[Editors’ note: what now follows is the decision letter after the authors resubmitted for further consideration.]

Thank you for resubmitting your work entitled "Age-dependent dormant resident progenitors are stimulated by injury to regenerate Purkinje neurons" for further consideration at *eLife*. Your revised article has been favorably evaluated by Sean Morrison as the Senior Editor and Mary Hatten as the Reviewing Editor.

You did a good job of addressing the key reviewer comments. The manuscript was particularly improved by the addition of data from confocal imaging of Ki67, pH3, and BrdU and retroviral labelling. You have provided clearer images of dividing immature Purkinje cells in response to diphtheria toxin ablation of ~50% of PCs. Inclusion of additional proliferation markers with confocal imaging show that FoxP2 cells can divide. Retroviral infection was also used to mark cells that had divided. These additional data provide enough evidence for publication, but we would ask you address a few additional details:

1) We are concerned with the use of the term "FoxP2 expressing progenitor (FEP)" as all PCs express FoxP2, mature or not. The cells you are studying are more clearly defined by their low levels of CALB1. Please consider that issue with respect to terminology.

2) Details are needed describing which retrovirus was used, the source, plasmid map, production etc. Some retroviruses do infect non-dividing cells so please include a discussion of why you believe your retrovirus only infects non-dividing cells.

3) In Figure 3B, Calb1 and FoxP2 appear to be mislabeled.

4) Figure 3—figure supplement 7 please provided separate images for each channel.

5) It appears that FoxP2 is partially outside the nucleus in BrdU^+^ cells (likely why the co-labeling was not obvious in non-confocal images). Is this the case? If so it would be interesting to discuss.

---

## [Author Response]

[Editors’ note: the author responses to the first round of peer review follow.]

As you can see from the reviews, while finding the manuscript of potential interest, all three reviewers have reservations about the major conclusion. Should you be able to address these critiques with new experiments, we would be happy to consider a new manuscript in the future.

The main concern of all the reviewers was that given how unusual our finding is that new Purkinje cells (PCs) can be generated after birth, we need to provide additional evidence for this finding. In addition, we should address whether FEPs are normally in the PC lineage. We have made the following major changes to our manuscript that we hope now convince the reviewers that new PCs are produced by proliferation of a transient progenitor in the PC lineage when neighbor PCs are killed:

1) We have performed GFP-retroviral labeling of proliferating FEPs following injury and show they become PCs. These results are presented in the revised Figure 2D-I.

2) As an approach to study the fate of FEPs during normal development, we quantified the number of FEPs and PCs from inbred newborn mice at postnatal day (P) 1, 5 and 30. We find that the number of PCs at P30 is similar to the number of PCs plus FEPs at P1, indicating that FEPs normally have a delay in differentiation until after birth. This result is incorporated into the revised Figure 3C and discussion of results.

3) We included confocal images of co-localization of EdU+ FEPs with additional proliferation markers, Ki67 and pH3 (Figure 3E-X).

4) We show that BrdU incorporation into PCs is not due to DNA damage following ablation by diphtheria toxin (Figure 2—figure supplement 3).

5) Throughout the paper, we improved the quality of our images and added orthogonal projections to show co-localization.

6) We have expanded our Discussion to include the reviewers’ suggestions about other possible regenerative mechanisms and the fate of FEPs.

Reviewer #1:

[…] Overall the paper is of high technical quality and findings are interesting. However, I have one major reservation in their interpretation that the proliferating cells that produce PCs represent "a previously unidentified and normally dormant progenitor population." This proposal raises a number of questions: What is the normal function of these FEPs (they surely cannot just prepare for DT/DTR-induced cell killing)? What is their fate if no injury occurs? If we do not know their normal fate, how can we claim that they are "unipotent"?

We agree with the reviewer that these are important questions, and have now addressed them by performing additional experiments to study the normal fate of FEPs. We reasoned that if FEPs are normally destined to become PCs but are delayed in their differentiation (expression of CALB1) until after birth, then the number of CALB1+ cells at P30 should be similar to the number of CALB1+ cells plus FEPs at P1. However, our original data was unable to address this question due to the high variation we observed between animals because: 1) Our ablation experiments were performed in mice with an outbred background, and 2) quantifications were performed on only 3 midline sections/brain. The variation at P30 was ~12.5% of the mean, which is higher than the total number of FEPs present at P1 (~10% of PCs) making it statistically impossible to address the fate of FEPs. To overcome this barrier, we analyzed the number of FEPs and CALB1+ PCs on every other slide of a half vermis (~25-28 sections) in inbred C57Bl/6 mice at P1, P5 and P30 (Figure 3C). Indeed, with this approach, the variation was only ~5% of the mean at P30. New data presented in Figure 3C shows a significant increase in the number of CALB1+ cells at P30 compared to P1 and the total number of FEPs plus PCs at P1 was not statistically different than the number of PCs at P30. This result is consistent with the reviewer’s suggestion below that FEPs normally differentiate into PCs after birth.

One alternative (and simpler) interpretation is that these FEPs are immature PCs that somehow can go back to cell cycle if their neighboring PCs are lost, and this ability diminishes as development proceeds (from P1 to P5). Indeed there is precedent in other neuronal types, such as horizontal cells in the retina, which can undergo cell division after fully differentiated in adults after loss of certain cell cycle inhibitor/tumor suppressor genes (Ajioka et al., 2007). It is possible that transient down-regulation of these cell cycle inhibitors in immature PCs, triggered by environmental signals due to loss of neighboring PCs, could cause immature PCs to re-enter the cell cycle. To me, this alternative developmental plasticity is no less interesting.

We fully agree with the reviewer’s hypothesis, and indeed our new data described above supports the idea. We have also performed immunofluorescent analysis on sections at P1 for KI67 and P27^kip1^, followed by quantification of fluorescent intensity of either marker in the nucleus of FEPs compared to CALB+ PCs. This data is represented in new Figure 3—figure supplement 7. Although both populations showed overall weaker staining for both markers compared to granule cell precursors, we observed that FEPs had significantly lower P27 levels and higher Ki67 levels compared to PCs at P1. We have removed statements regarding the unipotency of FEPs from our manuscript, since we do not have the tools (FoxP2-FlpoER mice) for inducible fate mapping. We have added a discussion of possible alternative developmental plasticity phenomena to our revised paper.

I have a few other technical critiques:1) It seems unusual that there are fewer tdT+ neurons in the Pcp2-Cre;Ai14;LSLDTR mouse following DT injection (see Figure 1J-M and compare Figure 1P and 1Q). Is Pcp2-Cre, which should be expressed in all normal PCs after P7, no longer active in these "new" PC replacement cells? If so, how do we know these are real and/or functioning PCs?

Thank you for pointing this out. All the reviewers had a similar expectation. In our hands, however, the recombination efficiency of Pcp2-Cre is highly variable and lower than previously reported. We have analyzed litters from different males, but were unable to achieve higher recombination.

We added a more extensive analysis of the recombination efficiency at P1, P5 and P30 to our revised manuscript. This analysis shows no increase in the recombination efficiency after P1 (Figure 1—figure supplement 1F). We also show that following ablation at P1, all of the TdT-expressing cells are cleared and replaced within a week after injury, and that of the ~50% newly regenerated PCs, ~50% turn on TdT+ with time. This percentage is similar to the 50% of cells that normally express TdT and therefore results in a reduction in the total number TdT+ cells at P30 in P1-*PC-DTR* mice, compared to controls. We have included this explanation in the new manuscript.

As for normal function, the lack of motor behavior phenotype in P1-*PC-DTR* mice provides functional evidence that the regenerated PCs integrate properly into the CB circuit. Furthermore, in order to study the morphology of the regenerated PCs in adult mice, we performed BrdU injections 10-12 hr after DT and analyzed them at P30 (Figure 2—figure supplement 2). We observed that the cell bodies and dendritic trees of BrdU^+^ PCs at P30 appear normal.

2) In the P5 killing experiment, presumably by then Pcp2-Cre is expressed in a larger fraction of PCs compared with P1 (it would be useful to quantify this as done in P1 mice). Can this contribute to the difference between P1 and P5 ablation?

As described above, we did not observe a significant increase in the recombination efficiency of Pcp2-Cre at P5 compared to P1. The numbers of TdT+ cells/section (hence the number of cells ablated) were similar at the two ages. Therefore, we do not think that this is a contributing factor to the phenotype observed in P5-*PC-DTR* mice. We have highlighted these results in the text in the relevant sections and also added a new graph to Figure 1—figure supplement 1F that shows the similar recombination efficiency at the two ages.

3) It appears that the authors deduce PC soma size from 14 µm sections. Given PC soma size is larger than 14 µm, this kind of analysis would produce a large number of half-cut, incomplete PC somal images, which might skew the conclusion?

We thank the reviewer for pointing out this detail that we did not fully describe. For our soma size analysis, we only quantified the cells that had a visible primary dendrite, as this should indicate that the section was taken from the largest region of the PC soma. However, in order to ensure we are not including any bias to our quantifications, we repeated the same quantification with sections that were either 14 µm or 35 µm thick from the same brain and compared the results. As can be seen in the graph below, we found the soma sizes were similar using both methods (**Figure 1**, t-test, P=0.13). Therefore, we are confident in our results and we did not replace the quantifications provided in the first version of the paper. We also revised the Methods section for paper to clearly describe our approach.

**Author response image 1. respfig1:** Soma size determination by quantifying thick and thin histological sections gave similar results.

Reviewer #2:

This manuscript describes regeneration of cerebellar Purkinje cells following diphtheria toxin injection at P1 but not at P5 and suggests that this regeneration is through cell division of a "previously unidentified progenitor population". Based on the data shown and previous studies of PC cell number, other possibilities exist to explain the findings. I am not convinced that the paper demonstrates generation of new Purkinje cells.The specific comments are:1) There is wide variability in the number of cells expressing Td-tomato/diphtheria toxin receptor at P1 (+/- 22%) which makes it difficult to know how many cells are dying. No counts of the number of cells expressing Td-tomato/diphtheria toxin are given for P5 mice. Presumably the percent of cells would be higher as Pcp2 is not fully expressed in all PCs at P1.

As described in detail above (reviewer # 1, answer 3), we have now included the P5 data and described that in our hands we do not observe an increase in the recombination efficiency (and number of TdT+ cell) at P1 compared to P5 (or P30)(revised Figure 1—figure supplement 1). Although the variation in our recombination efficiency was high, we were able to observe regeneration of PCs since all of the P30 P1-*PC-DTR* animals had a normal number of PCs, showing that whatever number of PCs was lost (30-70% of PCs) they were replaced. There was likely an effect of high recombination variation in P5-*PC-DTR* animals because the behavior analysis showed that not every P5-*PC-DTR* animal was abnormal. When we analyzed the correlation between latency to fall from the rod and number of PCs, we observed that animals with smallest numbers of PCs performed the worse.

2) The representative image of DT treated PCs at P30 (Figure 1M) clearly shows a large number of missing PCs which would normally form a continuous monolayer, suggesting that the number of PCs did not recover from cell death. Is this a representative image?

We apologize for the confusion. We have replaced the picture with a more representative one. Due to our section thickness (14µm) we consistently see holes in the PCL in controls and mutant mice, depending on where in the PC soma the section is cut. Because of this, we performed all our analyses on 3 midline sections/brain.

3) Throughout the figures, co-labeling of PC markers with proliferation markers is not convincing. Images showing definitive co-localization should be provided.4) In general, the quality of the images is not high and very few co-labeled cells are shown. It is not clear that some of the very few BrdU^+^ cells shown are not Bergmann glial cells. High quality images of BLBP stained BGs would be more convincing than the low power images provided. In general, higher magnification and resolution images would be potentially more convincing than the data shown.5) Other studies have shown BrdU labeling of PCs in the adult, suggesting that BrdU or Thy incorporation assays detect DNA replication as well as cell division. Again, the co-labeling experiments with Ki67 and BrdU are not convincing. H3 would have been stronger evidence for cell division.

In order to address these three similar points (3-5), we have added orthogonal projections from zstacks obtained by confocal microscopy to demonstrate better co-localization in our figures, in addition to showing examples of Ki67+ and phosho-H3+, EdU+ FEPs. Furthermore, as requested by reviewer #3, we have performed retroviral labeling of proliferating FEPs following ablation, further confirming that the PCs are replaced by a progenitor that proliferates (new Figures 2 and 3 and the related supplementary figures).

In addition to this newly presented evidence, an increase in the number of FEPs following DT at P1 further confirms that the regeneration is achieved by a proliferating progenitor (Figure 3D).

6) The authors suggest that Calb1+ PCs can develop within one day after a putative cell division. How do they imagine such cells would differentiate, i.e. extend dendrites into the field of parallel fibers? There are no images of the newly generated PCs that convince me that they are immature PCs.

We were also surprised that regeneration is so rapid, but as we now discuss in the paper, we think that the reason PCs are regenerated at P1 and not at P5 is in part because by P5 the circuit is too mature for the new PCs to integrate. At P1, there is little molecular layer (where the PC dendrites synapse with the parallel fibers of granule cells). In addition, at P1-2 PCs have not settled into a monolayer, and have not formed their unipolar dendrites (they are multi-polar). The PCs settle into a monolayer and become unipolar and form distinct dendrites around P3-P5.

Because CALB1+ PCs are immature at P1-P2, we were not able to distinguish whether FEPs have a more immature phenotype than PCs. As described above (response #1 to reviewer 1), we followed the PCs that incorporated BrdU after injury to P30, and found they develop normal dendritic trees, and our behavior study provides functional evidence that the regenerated PCs successfully integrate into the circuitry (Figure 2—figure supplement 2).

Finally, our new data represented in Figure 3C suggests that FEPs normally differentiate into CALB1+ PCs during the first week after birth, which shows that the CB circuitry is able to accommodate maturation of PCs soon after birth. We now discussed the implications of these results in our manuscript.

7) There is literature suggesting that PCs undergo programmed cell death. While no direct studies have shown this, Bcl overexpression mice and Bax knockout mice both have ~30% more PCs. The authors should discuss these findings and how it might affect their results. If an immature (CALB1-) pool of PCs persists until P5, this would provide an alternative view to the idea that new cells are generated from "unidentified precursors".

Although we were not able to observe clear cell death of PCs by TUNEL at P3 and P5 in wild type mice, our new data indicates there might be a small drop in the total number of cells (FEPs + PCs) at P5, compared to P1 (Figure 3C). Our unpublished results, combined with previous literature suggests that there are multiple mechanism to ensure the correct number of PCs are generated in newborn mice, including apoptosis in the embryo to scale PC numbers with respect to other components of the CB circuitry and FEPs which replace any PCs lost to injury around birth. As described in our responses to reviewer 1, we agree that the FEPs are likely immature PCs that delay their differentiation until after birth, and this is now discussed in the paper.

Reviewer #3:

In this manuscript, Baylin et al. examined injury induced regeneration pf Purkinjie cells in neonates and proposed a model that upon elimination of Purkinje cells, a previously unknown quiescent progenitor population can be rapidly activated to replenish the loss cells (within 24hrs), resulting in normal cerebellum development. Furthermore, this progenitor population lost its capacity around P5. While the topic is interesting, the result is not convincing. High doses of BrdU is known to label DNA repair, especially in the context of massive injury and the authors mostly used protein markers for cell type identification. The process occurs so quickly (within 24 hrs) and in the fetal tissue. The authors should be able to use slice imaging to observe the death and birth of Purkinje cells directly, to illuminate any doubt of their model. In addition, the authors can use onco-retroviruses for lineage tracing to label new neurons and observe distinct Purkinje cell morphology.

We thank the reviewer for the excellent suggested experiments to remove any doubt of our conclusions. We performed intracranial injections of a GFP-expressing retrovirus 12 hours after DT injection at P1 to P1-*PC-DTR* animals and controls. When we analyzed animals 3 weeks after retroviral infection, we indeed observed rare GFP-labeled PCs, and only in the P1-*PC-DTR* animals. In No DT controls only the expected granule cells and glial cells were labeled. These results thus show that following depletion, ablated PCs are replenished by a proliferating progenitor (Figure 2D-I). In order to determine whether the BrdU incorporation we observed is due to DNA repair induced by DT-mediated cell death or actually due to proliferation, we performed irradiation (IR) of pups and repeated the BrdU regimen we used for our experiments. These new results are presented in the new Figure 2—figure supplement 3. We observed that following 3 injections of BrdU after IR, many cells showed y-H2AX foci in their nucleus, including FOXP2+ cells, however, we could not detect BrdU incorporation in any of the cells. On the other hand, high levels of BrdU incorporation were observed in the FOXP2+ cells in P1-*PC-DTR* animals of the same cohort. No DT control animals showed no BrdU incorporation in FOXP2+ cells, consistent with our previous observation. Furthermore, the lack of BrdU incorporation in the ectopic TdT+ cells (Figure 2—figure supplement 1) and the increase in the number of FEPs (Figure 3D) 12 hours after DT injection also support our conclusion that BrdU incorporation is due to proliferation and not DNA repair.

We have also tried performing live imaging of P1.5 CB slices from P1-*PC-DTR* animals on a *FoxP2TdT* background, with and without PC ablation (DT in vivo). One of the problems we have encountered is that the dense packing of the PCs at P1.5 makes it impossible to observe single PC cell bodies with the cytoplasmic TdT in our system. Due to the inconclusive initial results we obtained, we have not included the slice-culturing experiments in the new paper.

[Editors' note: the author responses to the re-review follow.]

You did a good job of addressing the key reviewer comments. The manuscript was particularly improved by the addition of data from confocal imaging of Ki67, pH3, and BrdU and retroviral labelling. You have provided clearer images of dividing immature Purkinje cells in response to diphtheria toxin ablation of ~50% of PCs. Inclusion of additional proliferation markers with confocal imaging show that FoxP2 cells can divide. Retroviral infection was also used to mark cells that had divided. These additional data provide enough evidence for publication, but we would ask you address a few additional details:1) We are concerned with the use of the term "FoxP2 expressing progenitor (FEP)" as all PCs express FoxP2, mature or not. The cells you are studying are more clearly defined by their low levels of CALB1. Please consider that issue with respect to terminology.

We have changed the terminology used in our manuscript. In the modified version, we are referring to the CALB1-/low FOXP2-expressing PC progenitors as immature PCs or iPCs.

2) Details are needed describing which retrovirus was used, the source, plasmid map, production etc. Some retroviruses do infect non-dividing cells so please include a discussion of why you believe your retrovirus only infects non-dividing cells.

We have updated our manuscript with the relevant details about the retrovirus used. As noted in the acknowledgement in the paper, we obtained it as a gift from our colleague Songhai Shi. We have included a paper describing his use of the retrovirus: Yu et al., 2009 and a review by Connie Cepko of the use of retroviruses for labeling neural progenitors, as well as references to the origin of the retrovirus vector. The retrovirus is a replication-incompetent retrovirus (Moloney murine leukemia virus) that expresses a modified version of GFP (superfolding GFP instead of EGFP) that was produced in HEK293T cells using the pCMV-vsvg and pCMV-gp packaging plasmids. Dr. Shi’s lab, as well as others in the field of neurogenesis, have been using the retrovirus for their clonal analysis of proliferating neural progenitors in the mouse neocortex for years. Such studies require that the virus only infects dividing cells. This is also a known property of retroviruses, since they require breakdown of the nuclear envelope during mitosis for transduction. One additional note is that we only observed labeled PCs in mice that received diphtheria toxin injection at P1 and never in the controls.

3) In Figure 3B, Calb1 and FoxP2 appear to be mislabeled.

We apologize for the mistake. We corrected the labels.

4) Figure 3—figure supplement 7 please provided separate images for each channel.

We added the single channel images to Figure 3—figure supplement 7.

5) It appears that FoxP2 is partially outside the nucleus in BrdU^+^ cells (likely why the co-labeling was not obvious in non-confocal images). Is this the case? If so it would be interesting to discuss.

Thank you for noticing this interesting point. It was a consistent finding that FOXP2 and BrdU (also pH3) signals do not overlap well in the PC nucleus. However, both signals appear to be in the nucleus. We highlighted this observation in the main text (and Figure 3—figure supplement 6 for pH3). We think that the poorly understood nature of the large PC nuclei where the DNA may only be condensed enough in particular areas of the nucleus (perhaps heterochromatin) for the BrdU to be detected may play a role in this phenomenon in non-dividing PCs, and in dividing PCs FOXP2 protein likely undergoes dynamics changes in chromatin localization.